# Sustainable Tourism Destination Image Projection: The Inter-Influences between DMOs and Tourists

**Dan Zhu \*** , **Jiayi Wang and Meifang Wang**

Glorious Sun School of Business & Management, Donghua University, Shanghai 200051, China; 2201290@mail.dhu.edu.cn (J.W.); 160720221@mail.dhu.edu.cn (M.W.)
* Correspondence: zhudan@dhu.edu.cn

**Abstract:** With the development of the Web 2.0 era, tourists can freely publish their destination experiences through online travel notes. This enables tourists to become important agents to project tourism destination image (TDI), impacting destination-sustainable development. Previous studies have compared the difference in the images projected by destination management organizations (DMO) and tourists through their published content. However, fewer studies have been done to explore the inter-influences between them on the diachronic process of TDI construction. From the perspective of "circle of representation" this question is researched through a case study of Chiang Mai, Thailand, regarding the market of mainland Chinese tourists. Through interviews and the collection of microblogs from the Thailand National Tourism Bureau and tourists' travel notes from 2009 to 2021, we found that Chiang Mai has experienced four stages of TDI construction, during which the "Xiao Qingxin" image is evolutionally constructed and formed into the representation circle. The inter-influences between DMO and tourists, as well as the influencing factors in this process, are summarized. Our study supplements a dynamic diachronic analysis of TDI from the constructivism perspective. Relevant management and marketing applications for TDI and destination sustainability in the post-pandemic and Web 2.0 era are also provided.

**Keywords:** tourism destination image; construction; projected image; circle of representation; Chiang Mai; Chinese tourist; big data

## 1. Introduction

Tourism destination image (TDI) is a classical inquiry in tourism research [1,2]. A successful and positive image would promote tourists' visitation and benefit the sustainable development of the destination. During the past five decades, a cluster of studies has been researching TDI separately from both the projected image on the destination supply side and the perceived image generated from the tourist side [3–8].

However, in the Web 2.0 era when tourists have free and flexible access to publish their real destination experience, the traditional distinction between the projected image from the supply side and the perceived image from the demand side becomes blurred [1,9]. Potential tourists may simultaneously receive the projected TDI from the online generated content of destination management organizations (DMOs), mass media, and tourists, such as from microblogs, advertisements, movies and travelogues. Furthermore, a new cycle of TDI projection and perception begins after these new tourists visit the destination and publish travel notes. Hence, tourists also participate in the TDI projection, making the TDI construction more dynamically evolute, involving conflicting, negotiating, competing, or integrating processes between different sources of online destination information. Rather than a fixed or stable concept, TDI should be more perceived as a constructive and accumulative process [1].

In this context, it is essential to explore how a successful TDI is projected through an interactive perspective on the online content generated by the two major agents, DMOs and

tourists, and what crucial factors influence this process. Doing so will provide marketing and management implications for tourism destinations. Nevertheless, relevant empirical studies are lacking in the current literature.

By taking Chiang Mai, Thailand, as a case, this study aims to explore how its destination image is constructed in the market of mainland Chinese tourists during the year 2009 to 2021 through big data analysis. The paper is structured as follows: in the literature section, we review the research of TDI and summarize the research gap showing the lack of diachronic analysis of the TDI construction process, especially by investigating the inter-influences between DMO and tourists in TDI projection. Then we introduce the model of the "circle of representation" to construct our research framework. The findings portion follows the study context and methods sections. We will reveal that it generally experiences four stages to accumulate and develop a successful image of "Xiao Qingxin" for projection to the Chinese market. Finally, we present the discussion and conclusion sections. Sino-Thai cultural proximity, Thailand's economic reliance on Chinese outbound tourists, and the Web 2.0 environment are identified as three important factors influencing the dynamic development of TDI projection. Theoretical and practical contributions are also provided.

## 2. Literature Review

### 2.1. Tourism Destination Image

Dating back to 1975 when Hunt [10] firstly proposed the concept of "tourism destination image", it has received extensive research attention from multiple disciplines for nearly five decades. Most literature focuses on exploring tourists' perceived image, i.e., the mental construct of tourist's individual impressions of and ideas about a certain tourism destination [11]. A large quantity of work has been accumulated on TDI, including the examination of its conceptualization and dimensions [11–14], the formation process and influencing factors [9,13,15–21], assessments and measurements [14,20,22], and its impact on tourists' decision making and behavior [17,23–26]. Among these, there are some representative works inspiring our understanding of this concept. For example, Kislali et al. divide tourist perceived destination image into the primary and secondary image, with the former referring to a tourist's destination image after real visitation and the latter referring to the image generated from secondary sources of information beyond one's own experience [9]. The secondary image is further divided into organic and induced images according to whether the source of information is not directly aimed at promoting the destination (organic) or intentionally promoting this destination (induced) [15,27,28]. Lai and Li summarized the three classical models of the TDI structure. They are the three-dimensional model (from functional to psychological characteristics, from common to unique image attributes, and from individual attributes to holistic impressions) [29,30], the causal-networking model (mainly referring to the "cognitive-affective-conative model" [31], the "cognitive affective-overall model" [13]), and the core-peripheral model [12,32]. Even though there are continuous debates about TDI structuring, another group of scholars contend that TDI should be understood as a holistic construct or gestalt experience [9,33–36].

Besides the tourist's perceived image, a group of literature distinguishes TDI according to different subjects that give birth to relevant concepts such as residents' destination image [16] and the image projected mainly by DMOs [8,37,38]. Before the Web 2.0 era, the distinction between the projected and the perceived TDI is relatively clear, since the expense of publishing the printed brochures and guidebooks or shooting the official destination-marketing advertisements "imparts a certain degree of authority on the marketers' behalf" [1] (p. 222). Following this line of thought, many scholars are examining the impacts of destination marketing strategies or TDI projection on the tourists' image perception [27,39], or comparing the differences between the projected images from the destination supply side and the perceived images from the demand side [3,5,6,8,40].

However, in the increasingly developed Web 2.0 world, the boundary between the official projected TDI and tourists' perceived TDI becomes more and more blurred [1,41,42].

The tourism officials and organizations can utilize social media platforms to promote the destination image [43]. Individual tourists can also freely express and publish their evaluations of or visiting experiences to the destination, relying on multiple sources of online platforms such as social media and online travel agencies (OTA), creating User-Generated Content (UGC) [44]. Recent studies have found the obvious influence of UGC on the formation of perceived TDI among potential tourists [17,45–48]. In other words, UGC is increasingly implicated as an important alternative source away from traditional DMOs and destination marketers to project the TDI. Some studies even indicate possible inter-influences between UGC and the DMO-published content in cocreating a successful TDI in this globalized internet environment [49–51]. Generally, the study of TDI experiences a paradigm shift from a stable and fixed concept to a dynamic and accumulative process under the constructivism perspective, especially after stepping into the Web.2.0 era [9,52,53].

In this context, a new question emerges: How do the official DMOs and the tourists influence each other's TDI projections through their online generated content? Is it possible for DMOs to refer to the tourists' UGC when exploring a TDI that is attractive and sustainable enough to attract potential tourism markets? What are the crucial influencing factors in this process? The answers to these questions are important to the sustainable development of the tourism destination and would shed light on its marketing and management practices. Currently, relevant diachronic process analysis of the inter-influences between DMO and tourists concerning their online generated content to construct the projected TDI is lacking in literature.

*2.2. The Theoretical Perspective: "Circle of Representation"*

To research the DMO-tourist inter-influence on TDI projection, we take from the idea of "the hermeneutic circle of representation" as our theoretical framework. It was initially proposed by Urry [54,55] in his seminal book *The Tourist Gaze*, complemented by the work of Hall, Butler and Jenkins [56–58]. It describes the cyclic process of TDI representation in four stages [58]: The first is the stage of image projection. TDIs are projected by mass media collectively. The second is the stage of image perception. The projected TDIs are perceived by individuals and may inspire their visitation to the destination. The third stage is the visitation of tourism icons. At the destination, the tourists will probably visit the tourist icons or sites seen in the projected TDIs. The fourth stage is the reproduction of TDIs. Tourists visiting the tourism icons will photograph their experiences at the site as proof of their visit. These proofs are displayed and spread to others after the trip as another form of image projection. It begins the cycle again by influencing the perceived image of other potential tourists.

Jenkins proved the existence of the cyclic process of TDI representation through an empirical study comparing the images of travel brochures promoting Australia and the pictures taken by the Canadian backpackers to that country [58]. Based on his findings, he further conceptualizes "the circle" as an outward spiral anchored to a central tourism icon. Each whirl of the spiral adds new audiences to the TDI, and they may participate in updating the projection of this tourism icon through producing their own insights. In this way, the TDI is continuously in evolution, with circuitous cultural production and reproduction processes.

Jenkins' thoughts consider the agency of tourist audiences and possible inter-influences between different agents (e.g., DMO, mass media, tourists) to project TDI. Wijngaarden also proves the agency of tourists in TDI (re)production [59]. Therefore, Jenkins' developed model of "circle of representation" is more fitting to explain TDI in the Web 2.0 context, where tourists are empowered to use their agency to influence the TDI projection through UGC. On the other hand, the Web 2.0 era also creates a new environment where this model could be tested in the literal context. Through the internet and different social media sources, tourists have more choices for language use to represent their actual or imaginary impressions of the destination. Besides visual materials, such as photography and short videos, literal texts such as microblog texts, travel notes, travel tips, and comments are

also widely used [60]. Especially during the recent decade, many studies have proved the function of online literal texts in TDI representation [1,42,50,61]. Hence, it is fitting to adopt the "circle of representation" model to construct our empirical analysis into the online published text from both the DMO and tourist sides for their accumulative cocreation of the TDI in a typical case. Figure 1 shows our research framework. We will answer three questions: (1) What is the construction process of TDI? (2) What is the inter-influence between DMO and tourists in the TDI projection process? (3) What external factors influence this process? In this study, we mainly conceptualize the TDI as a holistic construct rather than as scattered imageries of the destination.

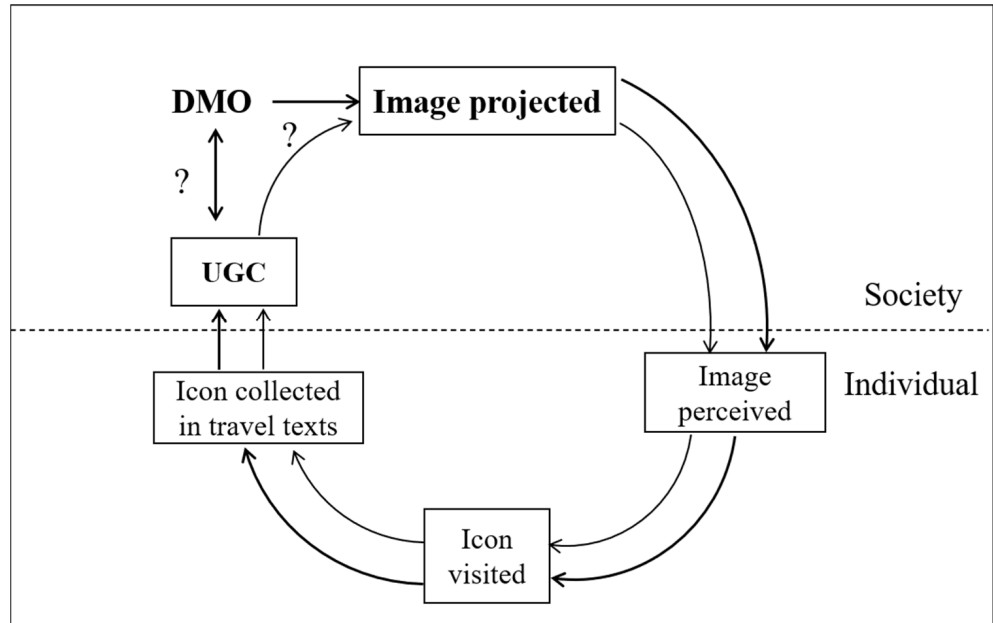

**Figure 1.** The research framework developed from the "circle of representation" theory.

## 3. Study Context

The TDI construction process promoting Chiang Mai, Thailand, to the mainland Chinese market is chosen as the study case. It is chosen for both substantive and practical considerations [62].

For the substantive considerations, Chiang Mai's tourism development within the Chinese market from 2009 to 2020 provides a typical case of how both tourist and DMO online-generated data are actively involved in the dynamic TDI projection. In 2009–2018, the compound annual growth rate (CAGR) of inbound tourists to Thailand was 10.1% compared to the 5.9% of CAGR during 1995–2008 [63]. Among the nations, China has always been Thailand's largest tourist-generating country. In 2019, Thailand received 39.916 million international tourists, and mainland Chinese tourists accounted for 27.6% of these [64].

From 2009 to 2012, tourist visitation to Chiang Mai, Thailand, was only on a small scale. Both the DMOs and tourists began to publish destination information online during those years. Large-scale Chinese mass tourism to Chiang Mai began in 2012, when a movie called *Lost in Thailand* became unexpectedly popular among the Chinese audience. Many scenes in this movie were shot in Chiang Mai, motivating many Chinese tourists to visit this province in Northern Thailand. This new tourist destination continued to develop for ten years, until the outbreak of COVID-19 in late 2019, turning Chiang Mai from an unknown location to one of the most popular Thai tourism destinations for mainland Chinese tourists. A famous image of "Xiao Qingxin" was successfully constructed for the Chinese market during this decade. Hence, it provides a clear time boundary for us to explore how a TDI

is born and gradually constructed into the representation circle, acting as a paradigmatic case [65].

For practical consideration, the first author has had an interest in Chinese outbound tourism to Thailand since 2012, obtaining relevant-themed funding from the Chinese government, and building a good academic relationship with Chiang Mai University, Thailand. These factors all provided accessibility for the authors to obtain both first-hand and second-hand data for researching the dynamic construction process of Chiang Mai's TDI for the Chinese tourist market.

## 4. Method

### 4.1. Data Cozllection

As this study aims to research the construction process of TDI, focusing on the DMO-tourist inter-influences in TDI projection, we adopt mixed methods, comprised of three stages, to collect and analyze data. In the pre-survey stage, the first author took an eight-day field trip in Chiang Mai, Thailand, from 8–15 August 2019. In this field trip, she visited the major tourist sites, including temples, the old town and streets, the university campus, night markets and shopping malls, the elephant camp, and the zipline in Chiang Mai. After obtaining informed consent, she conducted in-depth interviews with 22 Chinese tourists at the Chiang Mai International Airport. The first author asked the interviewees their primary impression of Chiang Mai before and after their visitation. Because of the time limit and the rate of tourists' refusal to grant informed consent, we did not collect much interview data at this stage. Data saturation was reached at the technical level rather than the conceptual level [66]. The first author transcribed and read the interview data each day after the field research. All the interviewees mentioned "Xiao Qingxin" in Chinese, to be the primary and satisfactory image of Chiang Mai, providing the basic assumption of our research.

In the formal data collection stage (from 20 January 2020 to 22 January 2022), we chose Sina Weibo as the source for DMO online data. Using its five offices in Beijing, Shanghai, Guangzhou, Chengdu, and Kunming, China, since 2011, the Thailand National Tourism Bureau has enrolled six legal Sina Weibo accounts to officially propagate Thai tourism in the Chinese market. These are the only official national departments in Thailand that publicly conduct online tourism marketing and cultural communication functions in China. Hence, the texts posted by these official accounts from 2011 to 2021 that included the keyword "Chiang Mai" (in Chinese) were searched, and we obtained 3997 posts from microblog text. For the tourists' part, as OTA websites focus on the tourism business, they provide public platforms for the tourists who have real destination experience to publish their travel notes using legal accounts. Comparatively, the identities of posters in Sina Weibo who publish Chiang Mai-related content are multiple, and it is difficult to identify which are tourists because most of the accounts use pseudonyms. Therefore, we choose the OTA websites as the data pool for tourists' UGC. We also searched the travel notes using the keyword of "Chiang Mai" (in Chinese) in five major Chinese OTA websites (Ctrip, Qunar, Tuniu, Mafengwo, Xinxin) from 2009 to 2019 and collected 1138 posts. To ensure accuracy, the second and third authors, respectively, conducted the data collection and then compared and shared the data with each other. The first author supervised and validated the entire process. Before 2009, most OTA websites did not provided a complete function for tourists to write and publish their travel notes. After 2019, as influenced by COVID-19, the Chinese government stopped its outbound tourism, hence the pause of tourists' travel notes in the OTA. This fact determined our starting and ending dates (2009–2019).

At the third stage (from 14 November 2021 to 24 January 2022), after obtaining informed consent, we interviewed three relevant informants, including local Thai officials and scholars, to help supplement the background information on the development of Chinese outbound tourism in Chiang Mai during 2009–2021.

All three stages of data collection help triangulate the data and improve the reliability of this study [67]. Specifically, the data from the on-site interviews with tourists inspired the basic assumption for our research question. In the second stage, big data is analyzed

to verify the assumption and answer the research questions. The third stage of interview data complements the storyline of the DMO-tourist inter-influences in the TDI construction process. Table 1 presents the profiles of all interviewees in this research.

**Table 1.** Profiles of interviewees.

| Chinese Tourists Interviewed in Chiang Mai, Thailand (1st Stage) | | | | | |
|---|---|---|---|---|---|
| **No.** | **Gender** | **Age** | **Origin (City, Province)** | **Profession** | **Length of Interviews** |
| C01 | Female | 25 | Shanghai | Foreign trade staff | 32 min |
| C02 | Male | 35 | Guangzhou, Guangdong | Unknown | 44 min |
| C03 | Male | 36 | Fuzhou, Fujian | Financial industry | 45 min |
| C04 | Male | 20 | Changsha, Hunan | University student | 26 min |
| C05 | Female | 28 | Suzhou, Jiangsu | Friends | 36 min |
| C06 | Female | 19 | Shanghai | University student | 45 min |
| C07 | Male | 27 | Nanjing, Jiangsu | Private business owner | 37 min |
| C08 | Female | 29 | Ningbo, Zhejiang | Bank employee | 30 min |
| C09 | Female | 50 | Suzhou, Jiangsu | Primary school teacher | 36 min |
| C10 | Female | 28 | Guiyang, Guizhou | Small entrepreneur | 25 min |
| C11 | Female | 26 | Guangxi | Civil servant | 18 min |
| C12 | Female | 34 | Guiyang, Guizhou | Vocational school teacher | 51 min |
| C13 | Male | 27 | Xishuangbanna, Yunnan | Real estate industry | 16 min |
| C14 | Female | 37 | Shenzhen, Guangdong | Unknown | 28 min |
| C15 | Male | 45 | Changzhou, Jiangsu | Private business owner | 47 min |
| C16 | Male | 36 | Shanghai | Accountant | 62 min |
| C17 | Female | 29 | Shanghai | Quality control personnel | 15 min |
| C18 | Female | 21 | Liaoning | University student | 27 min |
| C19 | Male | 33 | Wuhan, Hubei | Engineer | 15 min |
| C20 | Male | 44 | Zhuhai, Guangdong | Teacher | 10 min |
| C21 | Female | 21 | Beijing | University student | 42 min |
| C22 | Male | 29 | Shenyang, Liaoning | Engineer | 14 min |
| Informants interviewed online (3rd Stage) | | | | | |
| No. | Gender | Age | Origin (City, Province) | Profession | Length of Interview |
| O01 | Male | 35 | Surat Thani Province, Thailand | The Vice Director of the Thai-Chinese Belt and Road Research Center | 85 min |
| O02 | Female | 42 | Chiang Mai Province, Thailand | Tourism researcher at Chiang Mai University | 55 min |
| O03 | Male | 30 | Kunming, Yunnan | Management staff of the official Weibo of Thailand Tourism Bureau | About 60 min |

*4.2. Data Analysis*

The data analysis was conducted in four steps. First, the authors repeatedly read all the posted texts to get an overall idea of the development of Chiang Mai's projected TDI as constructed by the DMO and tourists. Corresponding with our interview data at the pre-survey stage, "Xiao Qingxin" is frequently mentioned to represent Chiang Mai in these texts. Second, with the help of the ROST Content Mining System 6.0, we conducted the content analysis. The ROST Content Mining System 6.0 is software used for analyzing big data in Chinese, developed by a research group led by Dr. Shen Yang at Wuhan University, China. It has been widely used in tourism studies for the calculation of word frequency, word cloud, sentiment, and social network analysis [60,68]. We calculated the word frequency of the travel note texts annually, with an eye on "Xiao Qingxin" for appearance time, word frequency, and its ranking in the high-frequency word list. We summarized its annual ranking change using a line chart to define the evolving pattern of the image "Xiao Qingxin" projected by Chinese tourists during 2009–2019. The same method was conducted for the DMO's data, and we compare the patterns between these

two. Combined with the interview data obtained at the third stage, we outline the sequence between the DMO and tourists, determining who first projected the "Xiao Qingxin" image and who was the follower. Third, we returned to the texts mentioning "Xiao Qingxin" and conducted a semiotic analysis to further obtain its connotations [58]. Following the steps of thematic analysis [69], we obtained four different themes of connotations for the image "Xiao Qingxin" and their evolutions throughout the years of research. Fourth, based on the findings of the previous three steps, we referred to the interview data again, further sorting out the timeline for Chinese outbound tourism development in Chiang Mai. After these four steps, we finally summed up the four stages of the evolution of Chiang Mai's TDI marketed to China, during which "Xiao Qingxin" was produced and co-projected by tourists and DMOs as a successful and sustainable TDI.

## 5. Findings

### 5.1. The Exploring Stage (before 2012): Blurred Image of Chiang Mai

Before 2012, Chiang Mai was unknown to most mass Chinese tourists, and it had not developed a clear destination image in the mainland Chinese market. The word "Xiao Qingxin" did not appeared in the tourists' travel notes or official microblogs before 2012. According to our annual word frequency analysis (2009–2019), hotel, temple, massage, cheap, elephant, night market, and service appear in the top 35 high-frequency word lists each year. Nevertheless, from 2009 to 2011, there are scattered images of Chiang Mai.

In 2011, three offices of the Thailand Tourism Bureau (Shanghai, Guangzhou, and Kunming) first opened their Weibo accounts. Chiang Mai's image is ambiguous in confusion with other adjacent regions such as the "Mekong River" "Golden Triangle" and "Chiang Rai" in the DMO's text. The top 35 high-frequency words in both the tourists' travel notes (2009–2011) and the Thai Tourism Bureau's microblogs (2011) are summarized in a comparative word cloud figure (Figure 2). We can see that only "hotel" "shopping" and "culture" are common between the tourists' and DMO's texts. In this stage, we could not find a high-frequency word in both texts that generalized the holistic image of Chiang Mai.

**Figure 2.** The official (**right**) and tourists (**left**)' word cloud during the exploring stage.

### 5.2. The Developing Stage (2012–2016): Tourists Generate the "Xiao Qingxin" Image and the DMO Follows

In 2012, a Chinese comedy movie *Lost in Thailand* first screened in mainland China on 12 December, 2012. It tells the story of the adventurous journey of three Chinese in Thailand, and was shot in Bangkok and Northern Thailand (mainly including Chiang Mai, Pai, and Chiang Rai). This movie made over RMB 1.2 billion and was seen by around 40 million Chinese viewers. Thus, Chiang Mai is well-known to the Chinese audience, and large groups of Chinese tourists come to visit this location. The film director was also received and thanked by Thai Prime Minister Yingluck Shinawatra in 2013 for his contribution to the promotion of Thai tourism [70].

In 2012, the Chinese catchword "Xiao Qingxin" firstly appeared in tourists' travel notes to generalize the scattered attractions into a holistic image. From then on, it continuously ranks in the top 35 high-frequency words listed in tourists' travel notes (Figure 3), and over 80% of the travel notes mention it each year.

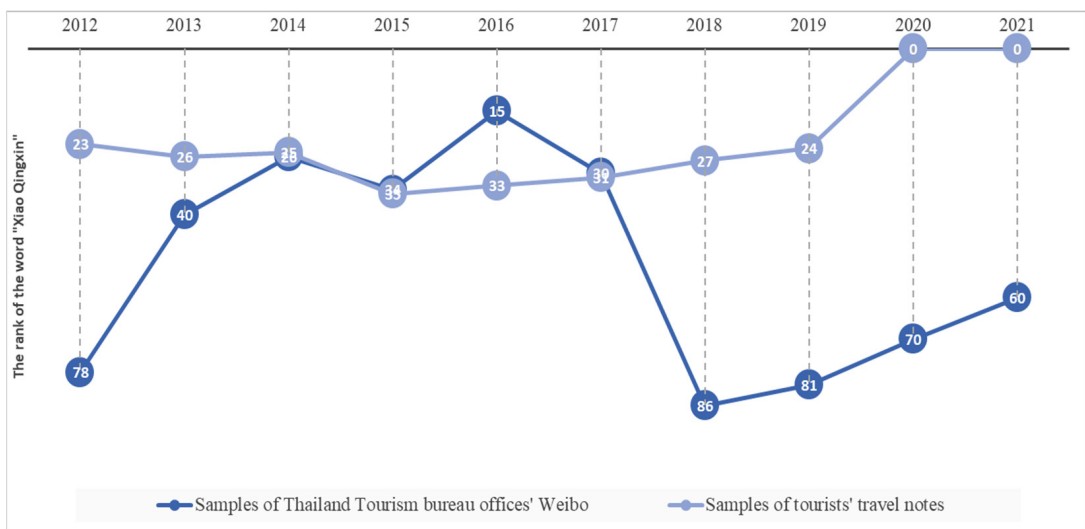

**Figure 3.** A line chart showing the word-frequency ranking of "Xiao Qingxin" in the annual sampling from both official and tourist descriptions.

According to our thematic analysis, it can be found that four different connotations of "Xiao Qingxin" are reflected in tourists' travel notes when describing Chiang Mai: (1) the style being literary, youthful, beauty-loving, and girlish; (2) the recreational environment/time being quiet, relaxing, warm, and of high life-quality; (3) the facilities, resources, and environment being pastoral, natural, and fresh; (4) an industrial or mental state being creative and exquisite, but not redundant. Table 2 presents the percentages of each connotation occurring annually in the tourists' text. During the first two years, "Xiao Qingxin" is mostly used by tourists to express the literary, youthful, beauty-loving, and girlish style in Chiang Mai. Chiang Mai serves as a typical background setting for them to self-present in this style, but its percentage decreases in the following years. In comparison, the percentage of the third connotation (the facilities, resources, and environment being pastoral, natural, and fresh) occurs in nearly 50% of all the travel note texts annually. This indicates that in the eyes of Chinese tourists, Chiang Mai earns the "Xiao Qingxin" title mainly because of its distinctive tropical climate, mountainous environments, and rich resources leading to the green, natural, fresh, and bright imageries.

Facing the large number of tourists coming all at once, both the DMOs and residents of Chiang Mai experienced a period of adaptation. At the very beginning, they did not prepare well and did not know what to do. O02's description of that period reflects this:

> Around 2012 to 2013, (Chinese) tourists came from the movie "Lost in Thailand". As I told you, so many Chinese came to Chiang Mai. First time we don't know each other, because you cannot speak Thai, and we cannot speak Chinese. We cannot communicate to each other. You don't know, because in Chiang Mai, we have only the Thai and English... We try to communicate, try to speak to you in Thai and in English, but you don't understand that.... So, I think the beginning is kind of complicated situation between us. (O02)

The same situation exists in the image projection of Chiang Mai by the Thai Tourism Bureau. As introduced before, its three official accounts were enrolled in 2011. According to our word frequency analysis (Figure 3), "Xiao Qingxin" did not appear in their microblogs published in 2011, but ranked number 78 on the list of high-frequency words in 2012. Its ranking generally continues increasing during 2012–2016. According to our interview with O03, an administrator managing the Thai Tourism Bureau's official microblog, the Thailand Tourism Bureau adopted this word from tourists and added it to their image projection for the Chinese market in 2012:

*"After 2012, with the popularity of "Lost in Thailand", Chiang Mai became more and more famous. "Xiao Qingxin" was first proposed by Chinese tourists visited Chiang Mai. While tourists share their experience on various online platforms, this title has gradually become popular, and has been widely adopted into the theme of the trip to Chiang Mai, Thailand". (O03)*

From 2012 to 2016, the six accounts of the Thai Tourism Bureau always retweet the Weibo posts from tourists, local Chinese residents of Chiang Mai, and other business accounts narrating the image of "Xiao Qingxin" (Figure 4). This can be thought of as the DMO's participation in projecting this image and pushing its circle of representation.

**Table 2.** Annual percentages of the different connotations of "Xiao Qingxin" in tourists' travel notes.

| No. | Connotations of "Xiao Qingxin" | 2012 (%) | 2013 (%) | 2014 (%) | 2015 (%) | 2016 (%) | 2017 (%) | 2018 (%) | 2019 (%) | Data Extracted from Tourists' Travel Notes |
|---|---|---|---|---|---|---|---|---|---|---|
| 1 | a style being literary, youthful, aesthetic, and girlish | 44.55 | 24.25 | 21.05 | 20.18 | 17.16 | 14.29 | 24.35 | 19.44 | "Chiang Mai left me the impression that whatever photos you took casually it should be 'Xiao Qingxin' literary style…" |
| 2 | the recreational environment/time being quiet, relaxing, warm, and of high life-quality | 3.96 | 8.58 | 7.66 | 8.26 | 9.70 | 14.29 | 1.74 | 6.94 | "I feel a sense of 'Xiao Qingxin.' The hotel is not big, but there are small decorations everywhere which is very warm to me." |
| 3 | facilities, resources and environment being pastoral, natural, and fresh | 39.60 | 40.90 | 42.14 | 45.30 | 45.37 | 52.78 | 47.96 | 49.15 | "The advantage of riding a bicycle is that you can enjoy the scenery along the way, which are all 'Xiao Qingxin' and pastoral. The weather is sunny, the air is fresh…" |
| 4 | an industrial or mental state being creative, exquisite, but not redundant | 11.88 | 22.39 | 23.44 | 25.69 | 22.39 | 21.43 | 24.35 | 33.33 | "Nimmana Haeminda Road gathered all kinds of bars, cafes, restaurants, clothing stores owned by designers, artists and entertainers, which is in strong sense of design and eye-catching. So, it's arguably the most 'Xiao Qingxin' street in Chiang Mai." |

From the perspective of the representation circle, the movie *Lost in Thailand* projected the primary images of Chiang Mai to Chinese audiences through its video frames. However, after tourists visit Chiang Mai, the original theme of "Jiong" (a Chinese character meaning "embarrassment") promoted in the movie is not taken by tourists to represent Chiang Mai. They spontaneously borrowed another catchword, "Xiao Qingxin" to summarize its image based on their actual experiences in Chiang Mai. By publishing travel notes, they projected this image, and it was further adopted by the official and commercial organizations in their publicity text targeted to the Chinese market. Hence, tourists initiated a new circle of representation, and updating the projection of Chiang Mai's image. The tourists' initiation of the image and its subsequent adoption by Thai officials are the unique features of this representation circle.

*5.3. The Adjusting Stage (2017–2019): DMO's Reformulation and Tourists' Renewing of "Xiao Qingxin"*

In this stage, the word frequency ranking of "Xiao Qingxin" on the DMO side decreases during these three years (see Figure 3). After the initial acceptance and adoption of this image developed by the tourist market, the Chinese tourists continued to visit Chiang Mai, forming a stable market. The DMO then begins to explore a localized formulation of Chiang Mai's image. The top 35 high-frequency words in the Thai Tourism Bureau's microblogs in the developing and diverging stages are summarized into a comparative word cloud (Figure 5). In Figure 5, "Loy Krathong "marathon" "Lanna" "music" "history" "creation" and "creative culture" became the new high-frequency words in the diverging

stage, indicating the officials' intentions to project the image of indigenous culture and festivals to represent Chiang Mai (Figure 5).

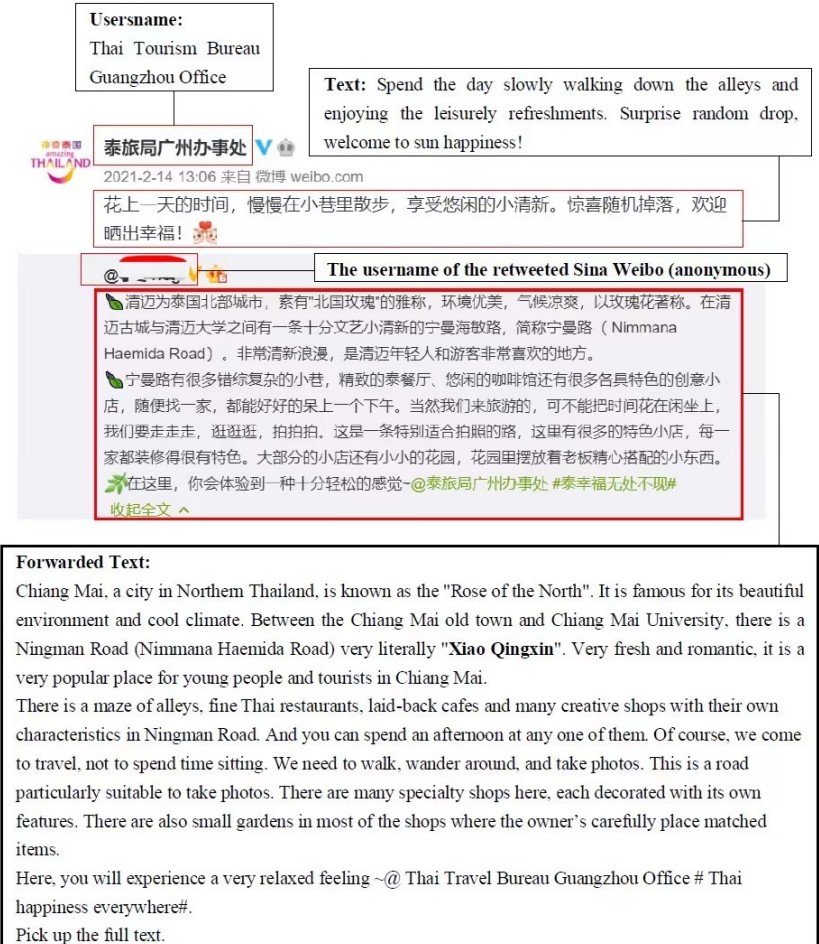

**Figure 4.** A retweet by the Guangzhou office, Thailand Tourism Bureau. Source: Sina Weibo (https://weibo.com/u/2428789347, accessed on 1 January 2022).

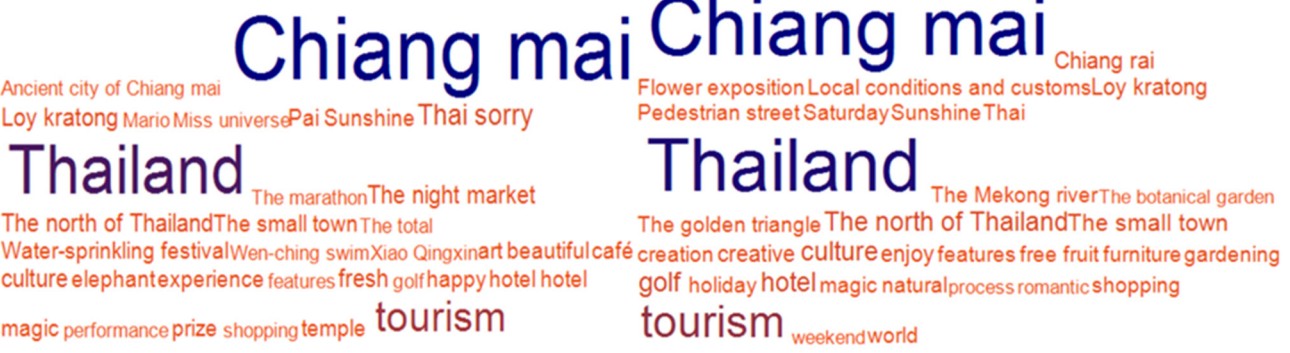

**Figure 5.** The word cloud comparison of the official microblogs in the developing stage (2012–2016, **left**) and diverging stage (2017–2019, **right**).

The interview with O03 proves this strategic change:

*"The main publicity focusses during 2017 to 2019: food is the mainstay supplemented by festivals, together with promoting the of historical sites. Festival promotion is focused on "Lanna culture", "Loy Krathong" and "Chiang Mai Creative Week". The goal is to create a holiday destination of food and culture. Over these years, there has been no change in*

*this strategy. The importance level from high to low is cuisine, festival culture, historical sites and natural attractions". (O03)*

In this period, the Thai Tourism Bureau intentionally reduced the use of "Xiao Qingxin" to reformulate Chiang Mai's image. They hoped to promote the holiday image by publishing microblogs relying on two major elements, food and culture.

This adjustment and relevant destination marketing subtly influences tourists' on-site experience, which can be seen in the tourists' travel notes. In this stage, "Xiao Qingxin" still appears in the top 35 high-frequency word lists for the tourists' text with an increasing rank (see Figure 3). According to our thematic analysis, tourist' texts about "Xiao Qingxin" still reflects four themes of connotations. However, its connotations are being renewed during this period. The percentage of the fourth connotation (an industrial or mental state being creative and exquisite, but not redundant) is annually increasing (see Table 2). This change reflects Chiang Mai's transition and upgrading from a destination for photographing and sightseeing to a holiday resort for experiencing creativity and mental replenishment in the gaze of the new-coming Chinese tourists. It indicates a correspondence from tourists with the DMOs to co-project the holiday image of Chiang Mai. Just as Jenkins has pointed out [58], although it is always anchored in a central tourism icon (the word of "Xiao Qingxin"), along with the publication of the travel experience of new audiences (the new Chinese tourists with fresh on-site destination experiences), its connotations are continuously updated in each whirl of the representation circle.

*5.4. The Returning Stage (from 2020-): DMO's Reprojection of "Xiao Qingxin"*

At the end of 2019, COVID-19 erupted, and tourism shut down. At present, Chinese outbound tourism has not yet recovered. From 2020 and 2021, the Thai Tourism Bureau kept continued posting new microblogs to project images of Chiang Mai to Chinese audiences, with a continued emphasis on food and culture. However, our word frequency analysis reveals that "Xiao Qingxin" ranks in the top 70 in the high-frequency word list each year, and its ranking continued to rise (Figure 3). This reflects the officials' reuse of "Xiao Qingxin" a familiar iconic word originally developed by Chinese tourists, to promote marketing and maintain Chinese interest in Chiang Mai. The strong desire to revive the Chinese tourist market can obviously be seen in O02′s words:

*"We welcome you because that Thailand, in Chiangmai, the main income came from tourism. So, we welcome every people who come to visit us. . . Now we miss you (Chinese) much. Please come to Thai". (O02)*

As the Deputy Director of the Thailand-China Belt and Road Research Center, National Research Institute of Thailand, O01 is quite familiar with both Thai and Chinese. He recognizes the significance of the Chinese word "Xiao Qingxin" in generalizing the image of Chiang Mai by reviewing the connections between these two:

*When you visit there (Chiang Mai), it is a "Lanna" image in clothes, architecture, in each aspect. Then if connecting Xiao Qingxin with Lanna Culture, can they be connected? In fact, there seems a certain correlation, because the culture, the architecture, the, including the speaking accent is very gentle. In all aspects, there is really a little bit of fit between these two. (O01)*

## 6. Discussion

Based on the empirical findings depicting the four stages of Chiang Mai's TDI evolution in the mainland Chinese market, three external factors particularly influencing the structure of this evolution are further summarized. First is the cultural neighborhood effect. In this case, Chiang Mai is located in Northern Thailand, which borders the Yunnan Province of China. Chinese tourists use a popular Chinese word, "Xiao Qingxin" to extract the characteristics of Chiang Mai, which to a certain extent echoes the local Lanna culture and can be accepted by local DMOs without inducing any cultural conflicts. Hence, our study enriches Kastenholz's [71] study about cultural proximity by elaborating its role in

bridging the gap in image projection between DMOs and tourists. Second is the economic reliance of the destination on the tourist-generating region. As revealed in the findings, Chinese tourists occupy the largest group of Chiang Mai's inbound tourist market. The strong intention to maintain and recover Chinese inbound tourism pushed the DMOs to (re)use the Chinese popular "Xiao Qingxin" image in their TDI projection. Third is the Web 2.0 environment that creates the possibility for inter-influences between different TDI-generated agents. The representation circle suggested by Urry was originally proposed in an era of printed travel brochures when visual text was more emphasized and tourists' agency to initiate the TDI projection received no real-time support. Beyond proving the previous studies [49–51] that the Web 2.0 era aids in realizing the coproduction and cocreation of TDI between tourists and DMOs, this study further points out the direct influence of tourists' UGC on the DMOs in TDI projection.

## 7. Conclusions

In regard to sustainable destination marketing and management, current TDI studies paid little research attention to the TDI construction process and the DMO-tourist inter-influences. This study takes Chiang Mai, Thailand, as a case and diachronically analyzes its TDI development for the mainland Chinese market. Through the perspective of the circle of representation and the analysis of multiple data sources, four stages of Chiang Mai's TDI projection to China are identified. First is the exploring stage (before 2012), when Chiang Mai is hidden beneath the images of its surrounding area and is unfamiliar to mass Chinese tourists. Second is the developing stage (2012–2016), when tourists are motivated by the movie *Lost in Thailand* to visit Chiang Mai and initiatively generate the "Xiao Qingxin" image. The DMO follows in projecting this image. Third is the adjusting stage (2017–2019) when the DMOs reformulate to promote Chiang Mai's holiday paradise image, focusing on food and culture, and the tourists correspondingly renew the connotation of the "Xiao Qingxin" image. Fourth is the returning stage (from 2020), when the DMOs intentionally reproject the "Xiao Qingxin" image to the Chinese market.

Our findings make some theoretical contributions. In the aspect of TDI knowledge accumulation, the current literature studies the function of the DMOs' and tourists' online generated content in TDI construction separately [38,46,47]. We supplement an empirical study into the dynamic evolution of TDI during which the possible DMO-tourist inter-influences are investigated over more than ten years. Our findings show that, through their UGC, the tourists directly influence the DMOs in their image projections. However, the influence of DMOs on tourists for image projection is much more indirect and subtle. As tourists publish their travel notes mainly based on their real destination experiences, they are seldomly influenced by the DMO's online published content; however, our big data analysis indicates that tourists' travel notes depicting their on-site destination experience will, to a certain extent, reflect the DMO's promoted images during the same time period.

In our study, we also enrich the "circle of representation" model. Unlike the model's original proposition that the destination organizations or mass media begin the representation circle of a certain TDI, thus attracting the tourists' participation, our case shows a different pattern. The mass media inspires tourists' travel, but the tourists themselves originally generate and project the new image of "Xiao Qingxin" for Chiang Mai, starting its representation circle. By reading travel notes online, potential tourists perceive this image and visit Chiang Mai. They visit the destination sites noted as "Xiao Qingxin" by previous tourists and reuse this word in their newly posted travel notes. It begins a new representation cycle of the image of "Xiao Qingxin." In this process, tourists' UGC influences the DMOs' TDI projection. The DMOs accept this popular image generated by tourists and participate in its representation circle. Corresponding to Jenkins' view, with the development of the destination, although as an icon word, "Xiao Qingxin"'s connotation may subtly change according to the evolution of tourists' on-site experiences. The cultural neighborhood effect, the destination's economic reliance on the tourist market, and the Web 2.0 context are further summarized as three crucial external factors influencing this model.

The updated structure of the representation circle model is shown in Figure 6. In this way, we also broaden the use of the representation circle model from analyzing only the visual text (e.g., pictures, photographs) to analyzing the written text as well. The updated model shows its explanatory potential, which could be tested in more empirical studies using different destination-market contexts.

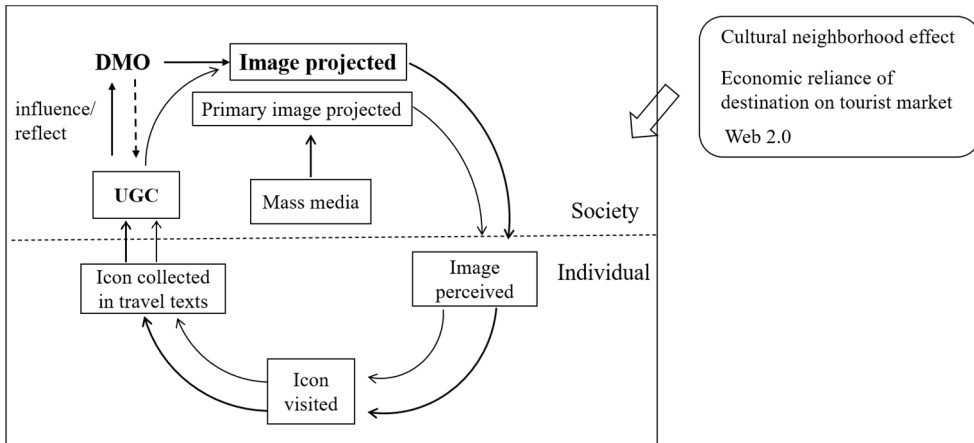

**Figure 6.** An updated model of the TDI representation circle in the Web 2.0 context.

Our findings provide some practical applications for DMOs in the sustainable management and marketing of TDI. First, it is important to take a strategic overview of the destination-market structure regarding their cultural and economic relationships. This forms the basis for making a TDI projection strategy which is stable and sustainable to the market. Second, as evidenced in this study, it is indicated to conduct a timely diachronic analysis into the UGC of tourists to discover some suitable concepts or icon words to generalize the TDI that suits the market's taste. Third, as an alternative marketing strategy, the DMOs could adjust their TDI projection by enriching or changing the connotations of a classical icon word, guiding the market interest without undermining its expectations.

Due to the limitations of time and accessibility, the interview sample size of the key informants is relatively small. Future studies can be conducted in two directions. First, it is significant to conduct more long-term longitudinal studies in the TDI evolution on typical cases. As many challenges exist for most tourism destinations in the world concerning tourism recovery during and after the pandemic period, the question remains concerning how DMOs will choose to deal with building a sustainable TDI when reencountering the sudden visitation of large groups of tourists. Second, it is important to make cross-cultural comparative studies about the DMO-tourist inter-influences in the TDI construction between different destinations and their tourist-generating regions. This would shed light on destination-sustainable development.

**Author Contributions:** Conceptualization, D.Z. and J.W.; methodology, D.Z.; software, J.W. and M.W.; validation, D.Z., J.W. and M.W.; formal analysis, D.Z. and J.W.; investigation, D.Z. and J.W.; resources, D.Z.; data curation, D.Z., J.W. and M.W.; writing—original draft preparation, D.Z.; writing—review and editing, D.Z. and J.W.; visualization, D.Z. and J.W.; supervision, D.Z.; funding acquisition, D.Z. All authors have read and agreed to the published version of the manuscript.

**Funding:** This research was funded by the National Natural Science Foundation of China, grant number 41901166, and the Department of Social Sciences, Ministry of Education of the P.R.C., grant number 19YJCZH276.

**Institutional Review Board Statement:** Ethical review and approval were waived for this study, because the Ethical Review Board of Donghua University do not require the full review process for the field work and data collection from adults who have adequate decision-making capacity to agree to participate.

**Informed Consent Statement:** Informed consent was obtained from all subjects involved in the study.

**Data Availability Statement:** Not applicable.

**Acknowledgments:** We would like to thank Yansha Li for her assistance in transcribing the interview data.

**Conflicts of Interest:** The authors declare no conflict of interest.

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
