# Peer review of "Sustainable Tourism Destination Image Projection: The Inter-Influences between DMOs and Tourists"

_sustainability, doi:10.3390/su14053073_

Round 1

Reviewer 1 Report

Thank you for the opportunity of reading and reviewing your interesting manuscript. The paper is well written, presenting the results of a well conducted qualitative research. The topic is interesting as well. I recommend enhancing the theoretical background by including in the analysis of the issue of social media in the context of destination image, e.g. Farhangi, S. and Alipour, H. Social Media as a Catalyst for the Enhancement of Destination Image: Evidence from a Mediterranean Destination with Political Conflict. Sustainability 2021, 13, 7276. https://doi.org/10.3390/su13137276; Sultan, M.T. et al. Social Media-Based Content towards Image Formation: A New Approach to the Selection of Sustainable Destinations. Sustainability 2021, 13, 4241. https://doi.org/10.3390/su13084241; Sun, W. et al. Examining Perceived and Projected Destination Image: A Social Media Content Analysis. Sustainability 2021, 13, 3354, https://doi.org/10.3390/su13063354. 

Other suggestions are as follows:

-separating the discussion and conclusions section into two different sections, i.e. Discussion and Conclusions

-including reference to the limitations of the study and possible application of the findings for wider context.    

Good luck!

Reviewer 2 Report

1) Dear authors, when working on a case study (single case), the justification of the case is very relevant. It is necessary to support the choice of the case (Chiang Mai, Thailand) with appropriate references.

2) Regarding the methodology, the authors must clarify how the interviews with tourists (22) and key informants (3) are complemented with the posted texts.

3) Also, what is the level of saturation in the responses of the 22 tourist interviews?

4) It is suggested to separate the conclusion from the discussion to highlight the contribution of new knowledge to this manuscript.
